# Analysis of Dynamic Plantar Pressure and Influence of Clinical-Functional Measures on Their Performance in Subjects with Bimalleolar Ankle Fracture at 6 and 12 Months Post-Surgery

**DOI:** 10.3390/s23083975

**Published:** 2023-04-13

**Authors:** Mario Fernández-Gorgojo, Diana Salas-Gómez, Pascual Sánchez-Juan, Esther Laguna-Bercero, María Isabel Pérez-Núñez

**Affiliations:** 1Movement Analysis Laboratory, Escuelas Universitarias Gimbernat (EUG), Physiotherapy School Cantabria, University of Cantabria, 39300 Torrelavega, Spain; 2Alzheimer’s Centre Reina Sofia-CIEN Foundation, 28031 Madrid, Spain; 3Neurodegenerative Disease Network Biomedical Research Center (CIBERNED), 28029 Madrid, Spain; 4Traumatology Service and Orthopedic Surgery, University Hospital “Marqués de Valdecilla” (UHMV), 39008 Santander, Spain

**Keywords:** malleolar fractures, dynamic plantar pressure, platform pressure, walking, functional scales, clinical measurement

## Abstract

Recovery after ankle fracture surgery can be slow and even present functional deficits in the long term, so it is essential to monitor the rehabilitation process objectively and detect which parameters are recovered earlier or later. The aim of this study was (1) to evaluate dynamic plantar pressure and functional status in patients with bimalleolar ankle fracture 6 and 12 months after surgery, and (2) to study their degree of correlation with previously collected clinical variables. Twenty-two subjects with bimalleolar ankle fractures and eleven healthy subjects were included in the study. Data collection was performed at 6 and 12 months after surgery and included clinical measurements (ankle dorsiflexion range of motion and bimalleolar/calf circumference), functional scales (AOFAS and OMAS), and dynamic plantar pressure analysis. The main results found in plantar pressure were a lower mean/peak plantar pressure, as well as a lower contact time at 6 and 12 months with respect to the healthy leg and control group and only the control group, respectively (effect size 0.63 ≤ d ≤ 0.97). Furthermore, in the ankle fracture group there is a moderate negative correlation (−0.435 ≤ r ≤ 0.674) between plantar pressures (average and peak) with bimalleolar and calf circumference. The AOFAS and OMAS scale scores increased at 12 months to 84.4 and 80.0 points, respectively. Despite the evident improvement one year after surgery, data collected using the pressure platform and functional scales suggest that recovery is not yet complete.

## 1. Introduction

Ankle fractures are very common and account for a high number of annual emergency department visits and a large socioeconomic impact [1,2]. Of all fractures, ankle fractures represent 9% of all bone fractures, being the largest of the load-bearing joints [3]. The most frequent types of ankle fractures are malleolar fractures, mainly unimalleolar and bimalleolar [4,5]. The incidence has been increasing over the last two decades to between 71 and 187 fractures per 100,000 people depending on age, sex, and geographic region [5,6,7]. The main cause is low-energy indirect trauma in elderly and middle-aged women, mostly associated with osteoporosis. In young people, mainly males, they are caused by high-energy trauma during sport [6]. Among the risk factors that can be associated with ankle fracture are the practice of sport itself [8], diseases such as osteoporosis, diabetes, or obesity [9], and lifestyle [10].

One in four malleolar fractures will require surgical intervention [3]; it is the treatment of choice if joint congruence cannot be restored conservatively. The most frequently used surgical treatment is open reduction with internal fixation (ORIF), the main purpose of which is to prevent post-traumatic arthritis and shorten immobilisation times [11].

The importance of the severity of the injury, the surgical intervention, and the immobilisation time implies significant physical and biomechanical alterations [12,13]. These alterations are manifested by increased pain, swelling, stiffness, soft tissue involvement, atrophy, and decreased muscle strength [14,15,16,17]. Consequently, these impairments indirectly impact functional activities such as gait [16,18,19] or limitations in work and leisure [20].

Several studies have reported short- and long-term outcomes after ankle surgery [21,22,23]. Beckenkamp et al. [23] concluded in their meta-analysis that 6 months after surgery, patients recover very slowly, and even at 24 months they do not reach full recovery. However, the assessment of disability after an ankle fracture is often based on scales such as the American Orthopedic Foot and Ankle Score (AOFAS) [24] or the Olerud–Molander Ankle Score (OMAS) [25], which incorporate different factors to describe function, alignment, and pain. Although these rating systems provide a simple way to assess the degree of disability and mobility of the patient, many of these factors are assessed subjectively.

Complementary to these functional scales, clinical parameters such as ankle dorsiflexion range of motion (ADF ROM), level of swelling, and muscle atrophy provide objective measures of clinical status at different stages of rehabilitation [18,26,27]. The recovery of these parameters after ankle surgery is critical and may play an important role in the full recovery of gait functionality [28] or balance [29].

Gait assessment has been widely used to characterise functional performance in patients who have sustained ankle injuries [19,27,28,30]; it is important for treatment decisions during rehabilitation [31]. In addition to kinematic characteristics and spatiotemporal gait parameters [32], analysis of plantar pressures allows us to identify subtle changes in foot loading that may go unnoticed on clinical examination [33]. This analysis enables us to quantify the magnitude and distribution of the force applied to the surface of the foot during gait. This is of great importance, as variations in pressure are associated with alterations in the moments acting on the foot and ankle joints [34,35]. In fact, in the clinical context, studies on subjects who have suffered an ankle injury find an association between the plantar pressure pattern and the characteristics of the injury itself [27,36,37,38]. Furthermore, the correlation between dynamic plantar pressure and clinical measures such as ADF ROM in subjects with osteoarthritis [36] or calf muscle atrophy in subjects with calcaneal fracture [37] has been reported in the literature. However, this correlation has not been studied in subjects with bimalleolar ankle fracture and how these clinical measures may influence plantar pressure performance.

The aim of this study was firstly, to analyse dynamic plantar pressure, as well as gait cadence and gait speed in subjects with bimalleolar ankle fractures at 6 and 12 months after surgery; secondly, to evaluate clinical variables and functions such as ADF ROM, bimalleolar/calf circumferences, and the AOFAS and OMAS scales over the same period of time; and finally, to study the association and potential influence of gait and clinical-functional variables with the results of the dynamic plantar pressure analysis.

## 2. Materials and Methods

### 2.1. Type of Study

This longitudinal research work was carried out on a sample of patients with bimalleolar ankle fractures at 6 and 12 months after surgery and a control group of healthy subjects. The evaluation was performed in a single session in the movement analysis laboratory of the Gimbernat-Cantabria University Schools (attached to the University of Cantabria).

### 2.2. Participants

The population consisted of 22 patients (10 women/12 men) who underwent surgery following bimalleolar ankle fractures at the Traumatology Unit of the Marqués de Valdecilla University Hospital (HUMV) in Santander. The surgical technique used was Open reduction with internal fixation (ORIF); the time elapsed from injury to surgery was 4.8 ± 7.6 days. After the immobilisation time (3.4 ± 1.2 weeks), progressive rehabilitation was performed for 6 weeks, consisting of passive stretching, kinesitherapy and ankle strengthening exercises. Once the orthopaedic surgical team authorised the progressive loading phase, participants undertook a balance, proprioception, and gait training programme lasting 13 ± 2.4 weeks. Rehabilitation sessions were held 5 days a week, lasting approximately 45 min and supervised by the physiotherapy service of the HUMV. 

Inclusion criteria were established as patients with 6 months of evolution after surgery and aged between 18 and 55 years. Patients with previous surgery on the lower limb, bilateral ankle involvement, functional alterations in the non-operated limb and neurological or rheumatic pathology were excluded.

Patients were selected through medical records registered at HUMV and with the collaboration of the traumatology team. After approval of the written informed consent by the Cantabrian Research Ethics Committee (CEIC) (reference: 2017.072), they were invited to participate by telephone or email, where they were informed of the objective of the study and the procedure to be followed for its implementation. Written informed consent was obtained from all study participants prior to data collection. 

In this study, we also had a control group (CG) consisting of 11 healthy subjects (6 females/5 males), comprised of university professors and employees who agreed to participate on a voluntary basis. These participants were chosen based on characteristics similar to the ankle fracture group (AFG) in age and sex. All of them were currently free of musculoskeletal pathology of the lower extremity, neurological or rheumatological problems, and with no history of such pathologies. 

### 2.3. Procedure

Data collection was carried out in a single individual visit at 6 and 12 months after surgery. During this time interval, the control group of healthy subjects was also assessed. Sociodemographic and clinical information concerning the surgery and the rehabilitation process was extracted from the medical records.

#### 2.3.1. Functional Scales

Firstly, the functional status of the patients was assessed using the American Orthopedic Foot and Ankle Society (AOFAS) Ankle-Hindfoot score [39] and the Olerud–Molander Ankle Score (OMAS) [25]. 

The AOFAS scale is a frequently used instrument to assess the outcome of ankle and hindfoot injuries. This clinical grading system, developed by Kitaoka et al. [24], combines subjective pain and function scores provided by the patient with objective scores based on the surgeon’s physical examination of the patient. The scale includes nine items that can be divided into three subscales (pain, function, and alignment). Pain consists of one item with a peak score of 40 points, indicating no pain. Function consists of seven items with a peak score of 50 points, indicating full function. Alignment consists of one item with a peak score of 10 points, indicating good alignment (performed by the surgeon). The peak score is 100 points, indicating no symptoms or deficiencies [40]. There is no consensus in the literature on how to categorise the scores obtained. Some authors suggest a classification as excellent (90–100), good (80–89), acceptable (60–79), or poor (0–59) [24].

The OMAS scale is a functional scale developed in 1984 and designed for the assessment of patient-reported symptoms following ankle fractures [25]. It includes nine questions referring to pain (0–25), stiffness (0–10), swelling (0–10), stair climbing (0–10), running (0–5), jumping (0–5), squatting (0–5), use of supports (0–10), and work/activity level (0–20). The score is calculated as the sum of each question from 0 to 100, this being the best possible score [41].

#### 2.3.2. Anthropometric Measurements

Next, following standardised anthropometric measurement recommendations [42], we recorded weight (kg), height (cm) and, bilaterally, limb length (cm) and ankle and calf bimalleolar circumference (cm).

Limb length was measured using a non-stretchable and flexible measuring tape with the greater hip trochanter and the floor as reference points. The participant was in a standing position with the limbs fully straight and the pelvis aligned in the horizontal plane. The measurement was performed three times to ensure that the greater trochanter reference was well located.

The calf and bimalleolar circumference were measured in the same way as for limb length, with the participants in a standing position and using an inelastic tape measure. The calf circumference was measured 20 cm superior to the external malleolus of the ankle. For the bimalleolar circumference, the uppermost part of both malleoli was taken as a reference. Three measurements were taken at each circumference, and the average was calculated.

#### 2.3.3. Ankle Dorsiflexion Range of Movement

ADF ROM (degrees) was assessed by means of a digital inclinometer (Acumar, Lafayette Instrument, Lagatette, IN, USA) in an active and loaded manner following the weight-bearing lunge test. This model allows a more functional active and loaded assessment for activities of daily living such as walking, running, or stair climbing. In addition, previous studies report superior inter-examiner reliability in healthy populations (0.93 ≤ ICC ≤ 0.96) [43] and with ankle fracture (0.90 ≤ ICC ≤ 0.99) [44] compared to the “no load” position (0.32 ≤ ICC ≤ 0.72) [45]. The starting point for the measurement was with the ankle at 90° and the inclinometer resting below the tibial tuberosity. With the subject barefoot, at a distance of 30 cm from the wall and the knee aligned with the second toe, the subject was actively brought to the limit of dorsal flexion. Prior to the measurements, the test was performed to familiarise the subject with the movement. Finally, three measurements were taken, and the two most similar values were averaged. 

#### 2.3.4. Gait Analysis

The gait analysis was performed with the subject barefoot on a walkway 8 m long and 2.5 m wide where he/she had to walk four laps (32 m) at their normal walking speed. We considered normal speed to be the speed previously preferred by each subject after a brief trial at different speeds, following the recommendations of some authors for the analysis of walking on flat terrain [46]. Two valid trials were performed for each participant. For gait analysis, a wireless inertial sensor system BTS G-WALK (BTS Bioengineering S.p.A. Italy) weighing 37 g and measuring 70 × 40 × 18 was used, placed by means of a semi-elastic belt at the level of the first sacral vertebra (S1). The methodology and complete evaluation of the spatiotemporal gait parameters was previously published by our research group [28]. The gait variables used for the present article were cadence (step/min) and gait speed (m/s).

#### 2.3.5. Dynamic Plantar Pressure

Dynamic plantar pressure was recorded using a single BTS P-WALK pressure platform (BTS Bioengineering S.p.A. Italy), with dimensions 640 mm × 740 mm × 8 mm and an acquisition frequency of 250 Hz. Data transmission to the computer is via USB2, where specific software (BTS G-Studio) processes the information received. The software itself identifies each step performed, and in the processing, the researchers themselves (M.F.-G. and D.S.-G.) take care of eliminating incorrect or incomplete steps. This pressure platform has been widely studied in healthy people and people with different pathologies, showing a moderate–high reliability (ICC > 0.7) in its records [47,48,49,50].

For data collection, the pressure platform was placed in the intermediate part (4 m) of the walkway. In this way, for step standardisation, we ensured that at least three steps were taken before stepping on the plate. Five steps were collected with each limb during a continuous gait cycle.

The variables recorded using the pressure platform were: weight-normalised peak plantar pressure (PPP) (kPa)—peak pressure recorded from the total number of steps taken during the gait; weight-normalised average plantar pressure (APP) (kPa)—average of the pressures recorded from the total number of steps taken during the gait; contact time (CT) (ms)—time elapsed from the start of contact to the takeoff of each step. 

### 2.4. Statistical Analysis

Firstly, participants in the ankle fracture group (AFG) were classified according to their operated (OA) and non-operated (NOA) ankle. For the control group (CG), the dominant limb was taken as the reference. Sociodemographic, clinical, and functional variables were described. For categorical variables, percentages with their corresponding 95% confidence intervals (95% CI) were estimated, and for continuous variables, means with their standard deviation were estimated. The Shapiro–Wilk test was performed to analyse the normality of the variables. 

In the AFG, the results of dynamic plantar pressure, gait parameters, clinical measurements, and functional scales were analysed in each of the two ankles (OA/NOA) at 6 and 12 months, as well as the intra-group difference of these variables in that time range. The mean difference between them was obtained using the Student’s t-test for paired samples, and the mean differences between groups (AFG/CG) at 6 and 12 months were obtained using the Student’s t-test for independent samples. The effect size was also calculated using *Cohen’s d*, whose values are quantified as follows: 0.2 small, 0.5 medium, and 0.8 large [51]. Previously, using GPower 3.1 statistical software and based on the difference between the two ankles (OA/NOA), we calculated for our ankle fracture sample (n = 22) an effect size of *d* = 0.62 (1 − *β* = 80%; α = 0.05) necessary for the test to be sensitive to detect a clinically relevant change. Similarly, for the between-group difference (AFG n = 22; CG n = 11), we calculated a necessary effect size of *d* = 0.93 (1 − *β* = 80%; α = 0.05) [52].

The degree of relationship in AFG of dynamic plantar pressure with clinical measurements, gait parameters, and functional scales was analysed using Pearson’s correlation coefficient (r). A multiple linear regression analysis (r^2^) was then performed with the variables that showed a significant correlation. Previously, to guarantee the validity of the regression model, we ensured that the assumptions of linearity, independence, normality, homoscedasticity, and non-collinearity were met. The model was completed with statistical power (1 − *β*) and effect size (*f_2_*), calculated using *R*^2^/1 − *R*^2^ and whose values are quantified as 0.02 small, 0.15 medium, and 0.35 large [53]. Statistical analysis of the data was performed using SPSS software (Statistical Product and Service Solutions IBM SPSS Statistics 25.0; 2017) and Excel for Mac (Microsoft 365; version: 16.51).

## 3. Results

Twenty-two patients (10 females and 12 males) with bimalleolar ankle fracture and 6 months after surgical intervention participated in the present study. The mean age was 43.5 ± 10.2 years, with ages ranging from 21 to 55 years. One patient did not present at the 12-month assessment due to personal issues unrelated to the study. The rest of the patients completed the two assessments without suffering any complication in the fracture, or associated pathology that could bias the results. Eleven healthy subjects (6 women and 5 men) with a mean age of 39.9 ± 8.6 formed part of the CG. Table 1 describes the demographic and anthropometric characteristics of both groups, as well as the functional status at 6 months of the AFG.

### 3.1. Differences in AFG between the OA and NOA at 6 Months and Compared to the CG

At 6 months after surgery, we found a significant difference between the operated and non-operated ankle in all clinical measures, with an effect size (*d*) in the range of 0.78 ≤ *d* ≤ 2.31. When we compare these clinical measures with CG, only the differences are significant in bimalleolar circumference (3.2 cm; *d* = 1.64) and ADF ROM (−19.1°; *d* = 2.71) (Table 2).

In relation to the results obtained for dynamic plantar pressure, we found differences in AFG between both ankles and a moderate effect size in PPP (−0.04 kPa/kg; *d* = 0.57), APP (−0.02 kPa/kg; *d* = 0.56), and TC (−24.8 ms; *d* = 0.72). These differences in dynamic plantar pressure are also significant when compared to the CG dominant limb. Specifically, we found differences in PPP (−0.32 kPa/kg; *d* = 0.69), APP (−0.16 kPa/kg; *d* = 0.63), and CT (128.8 ms; *d* = −0.97) (Table 3).

### 3.2. Differences in AFG between the OA and NOA at 12 Months and Compared to the CG

As was the case at 6 months after surgery, at 12 months the differences in clinical measurements between the two ankles in AFG remained significant (0.53 ≤ *d* ≤ 1.18). The same was found for comparison with the CG, where bimalleolar circumference (3.0 cm; *d* = 1.54) and ADF ROM (−12.2°; *d* = 1.48) were still the variables with the largest differences (Table 4).

With dynamic plantar pressure, only the differences were significant in the comparison of the OA of the AFG versus the dominant limb of the CG, with an effect size in the range of 0.71 ≤ *d* ≤ 0.73. Furthermore, both cadence (−7.3 steps/min; *d* = 1.06) and speed (−0.17 m/s; *d* = 1.21) showed significant differences between both groups (Table 5).

### 3.3. Differences in AFG between 6 and 12 Months

Regarding the differences obtained between 6 and 12 months in the clinical measures, they were significant only in the ADF ROM (−6.8°; *d* = 0.73). The AOFAS and OMAS scale scores also increased significantly by 10.8 (*p* < 0.001; *d* = 1.15) and 22.7 (*p* < 0.001; *d* = 1.73) points, respectively. Finally, in gait parameters, we found significant differences in cadence (−6.5 steps/min; *d* = 0.72) and speed (−0.07 m/s; *d* = 1.06), as well as in PPP (0.05 kPa/kg; *d* = 0.64) and CT (56.2 ms; *d* = 0.56) (Table 6).

### 3.4. Correlation of Dynamic Plantar Pressure with Gait Parameters, Clinical Measures, and Functional Scales

Correlation analysis of dynamic plantar pressure with gait parameters, clinical measures, and functional scales at 6 and 12 months after surgery are described in Table 7. Multiple linear regression analysis with the variables that obtained a significant correlation at 6 months showed that CT increases with decreasing cadence *r* = 0.81 and speed *r* = 0.81 (*F* (1, 21) = 34.5; Δ*r*^2^ = 0.76; *p* < 0.05), PPP increases with decreasing bimalleolar circumference *r* = 0.58 (*F* (1, 21) = 9.5, Δ*r*^2^ = 0.30 *p* < 0.001), and APP increases with decreasing bimalleolar circumference *r* = 0.52 (*F* (1, 21) = 7.5; Δ*r*^2^ = 0.23 *p* < 0.05).

The results of the multiple linear regression analysis at 12 months were similar to those obtained at 6 months for the same predictor and outcome variables. In particular, bimalleolar circumference, cadence, and speed can predict the outcome of plantar pressure dynamics by 30–67% (Appendix A).

## 4. Discussion

The purpose of this study was to identify clinical and functional limitations in subjects with bimalleolar ankle fractures at 6 and 12 months after surgery. In order to quantify these limitations, we combined objective tests such as dynamic plantar pressure during gait analysis and clinical measures, together with functional scales. We compared the variables studied with a CG and evaluated the degree of correlation between these variables in the AFG, as well as their clinical implication. The results obtained in the dynamic plantar pressure analysis of the AFG (OA vs. NOA) and compared with the CG showed a decrease in plantar pressure (peak and mean) and an increase in contact time in the operated ankle. In addition, during gait analysis the cadence and gait speed of the AFG was significantly lower and clinically relevant when compared to the CG. Our research group previously published the full results of the study of spatiotemporal gait parameters and their relationship to clinical-functional measures at 6 months post-surgery [28]. To our knowledge, we are not aware of any study analysing the dynamics of plantar pressures in subjects with bimalleolar ankle fractures 6 months after surgery. A study by Zhu et al. [27] in twelve subjects after trimalleolar ankle fracture found similar results to ours in the dynamics of plantar pressures. Specifically, PPP was lower in the 3rd to 5th toe area when comparing OA/NOA and in the total plantar area when compared to a group of healthy subjects. However, differing from our findings, they found no difference in the CT between the OA and HA, but compared to the healthy group (189.86 ms; *p* = 0.003). These differences found in CT compared to our work could be justified by the type of fracture, the time of measurement (4.5 months after surgery) or the significantly lower gait speed (0.65 m/s vs. 0.96 m/s). The results found in the dynamics of plantar pressures show the asymmetry between the two ankles and could be interpreted as a protective factor when walking. Studies carried out in different populations after ankle surgery found similar results in the study of plantar pressures [38,54,55]. In relation to walking speed, studies in healthy subjects show a positive linear relationship with PPP [56]. In the absence of studies in the ankle fracture population that relate gait speed to dynamic plantar pressure, this relationship could be extrapolated to our study population. However, in our investigation, we did not find a correlation of plantar pressures with speed, but we did find a negative correlation with ADF ROM and cadence with CT. Furthermore, speed and cadence were found to be the main predictor variables that conditioned the CT results by 76%. This influence of speed and cadence on CT could easily be predicted by fear of stepping with an injured ankle, pain, or ankle instability [16,27].

ADF ROM and ankle strength are the most studied clinical variables in the population that has suffered an ankle fracture [57]. Complementary to these clinical measures, both bimalleolar circumference and calf circumference help us to estimate the degree of swelling and muscle atrophy after an ankle injury [27,37,58,59]. Of note in our study are the differences found between the OA and NOA in ADF ROM (−12.70; *p* < 0.001), bimalleolar circumference (1 cm; *p* < 0.001), and calf circumference (−1.3 cm; *p* < 0.001). Furthermore, we found a negative correlation of plantar pressure (average and peak) with bimalleolar and calf circumferences, with bimalleolar circumference being the main predictor variable conditioning 23–30% of plantar pressure scores. These differences in clinical measurements are consistent with those reported in the literature in subjects with unimalleolar [14], bimalleolar [22], trimalleolar [27], or calcaneal fractures [37]; however, they do not study the correlation of these clinical measurements with plantar pressures. Adhesions and soft tissue involvement following injury and surgery lead to increased stiffness, decreased ankle range of motion, and calf muscle atrophy. As a consequence of these alterations, their influence on plantar pressures could be justified, just as it occurs during walking [28].

Assessment of plantar pressure dynamics one year after surgery helps us to identify subtle differences in gait that may go unnoticed visually. In our investigation, in the AFG we found no differences between the two limbs in PPP, APP, and CT. However, when compared to the CG, the differences were significant in all three parameters. A study by Becker et al. [60] in forty subjects evaluated at 18 months after ankle surgery found significant differences in plantar pressures when compared to a group of healthy subjects. However, in contrast to our findings, they found an asymmetry between the two limbs of the AFG. Furthermore, the same authors identified an association between poorer functional status and lower values of plantar pressures. Hirschmüller et al. [37], in a group of sixty patients with intra-articular calcaneal fractures, found at 12 months after surgery a decrease in PPP in the hindfoot and an increase in the midfoot and lateral forefoot. However, in contrast to the work of Becker et al., there is no relationship between the functional scales and the dynamics of plantar pressures. In our work, we also found no such association with the AOFAS and OMAS scales; in contrast, we found a negative correlation between the circumferences (bimalleolar and calf) with PPP (r = −0.67; r = −0.61) and APP (r = −0.59; r = −0.52). This correlation suggests that the involvement of these clinical parameters strongly influences the pre-injury gait status. In particular, bimalleolar circumference could explain up to 43% of the results of plantar pressures. 

Gait speed and gait cadence, as was the case at 6 months, showed significantly lower and clinically relevant values (1.06 ≤ *d* ≤ 1.21) compared to the control group. Moreover, both parameters still conditioned the outcome of the CT in 67%. In agreement with our results, Losch et al. [61], in their study carried out with a sample of twenty subjects with ankle fractures (twelve bimalleolar) and at 12 months post-surgery, found a significant difference in walking speed when compared with a group of healthy subjects. On the other hand, van Hoeve et al. [26] found similar results at 18 months post-surgery in thirty-three subjects with unstable ankle fractures (eleven bimalleolar). In contrast, Wang et al. [30], in a group of eighteen subjects at 13 months after ankle surgery, found no such differences in gait speed when compared to healthy subjects. These contradictory results with respect to ours and other authors could be explained by the type of fracture (twelve subjects with a unimalleolar fracture and six with a trimalleolar fracture) or the protocol used in the gait analysis. 

Improved ankle mobility is a key determinant in the full recovery of the affected limb. Some authors conclude that a decrease in ADF ROM of more than 4.5° in the push-off phase is clinically relevant [26]. In our sample, we could not analyse the ankle range of motion during gait, but in the weight-bearing lunge test, we found a difference in ADF ROM of −7.4° (d = 1.18; *p* < 0.001) between both ankles of the AFG and −12.2° (d = 1.48; *p* < 0.001) compared to the CG. Nilsson et al. [22], in their retrospective study conducted in fifty-four subjects 14 months after ankle fracture surgery, found a difference in ADF ROM of −5.7° (*p* < 0.001) between the OA and NOA. In addition, they detected a significant difference in bimalleolar circumference of 1 cm (*p* < 0.001). In contrast to our findings, they found no difference in calf circumference. These results seem to indicate that the degree of ankle swelling 12 months after surgery, together with the intrinsic characteristics of the fracture, would largely explain the increase in ankle stiffness and thus its impact on the functionality of the lower extremity. In this regard, it has been studied that a minimum of 30° of dorsiflexion under load is necessary to be able to perform tasks such as descending stairs, squatting, or getting into a chair without problems [22].

In general, improvement in AFG between 6 and 12 months was evident in gait speed and cadence, dynamic plantar pressures, clinical measures, and functional scales. However, there are certain parameters whose improvement was not clinically relevant. Specifically, with regard to plantar pressures, the APP (−0.01 kPa/kg; *p* = 0.193; d = 0.26) was similar at 6 and 12 months. Despite the slight improvement in PPP (−0.05 kPa/kg; *p* = 0.023; d = 0.64) and CT (56.20 ms; *p* = 0.024; d = 0.56), there seems to be an attempt to reduce weight bearing on the affected limb due to pain or fear of stepping. Some authors state that it is impossible to determine whether the changes in plantar pressure distribution are external consequences of intra-articular biomechanical alterations, in the sense of compensatory mechanisms, or an acquired pattern to alleviate pain [60]. Another explanation would be given by limitations in ankle ADF ROM and/or lack of strength of the foot and limb musculature [26].

Regarding the results of clinical measurements 6 months after the first measurement, we found a significant improvement of the ADF ROM (−6.8°; *p* = 0.003; d = 0.73). However, the improvement in bimalleolar circumferences (0.2 cm; *p* = 0.570; d = 0.12) and calf (−0.6 cm; *p* = 0.160; d = 0.32) was very small and clinically not relevant. Nilsson et al. [62] compared in one hundred and ten patients operated after ankle fracture two types of interventions (specific training programme and conventional physiotherapy) performed for 12 weeks. Among their results, they found no significant differences between groups in ADF ROM at 6 and 12 months. However, both groups improved by 2° from their initial values, although this improvement could be considered insignificant.

The improvement that we were able to see in the clinical assessment and certain gait parameters is also reflected in an increase in the AOFAS and OMAS scale scores. Specifically, the AOFAS scale reached 84.4 points compared to 73.6 points 6 months after surgery. This result is similar to that found by other authors at 12 and 18 months’ follow-up, with scores varying between 84 and 90 points [13,26]. At 12 months after surgery, pain is mild and occasional, the ADF ROM is not complete, and restrictions are practically limited to sports and recreational activity. To date, the minimum clinically relevant difference for ankle injuries is not clearly specified, although some authors propose a difference of 6 points on the AOFAS scale [63]. Therefore, given the increase of almost 11 points between 6 and 12 months after surgery in our results, it can be considered clinically relevant. 

The OMAS scale score also improved considerably, from 57.3 points at 6 months to 80 points at 12 months. Our findings are in line with those reported by several authors one year after surgery with OMAS scores between 75 and 85 points [22,30]. Among the results obtained, they highlight the residual pain that patients still experience when performing activities such as walking, climbing stairs, or squatting. In addition, only 19% of patients reported full recovery, with limitations in sport, activities of daily living, or work [22]. 

The present study has certain limitations. Firstly, we have a relatively small sample, although, for the identification of clinically relevant intra- and inter-group differences, we found a moderate-to-large effect size (AFG d = 0.62; AFG/CG d = 0.93) with a power (1-β) of 80% and α = 0.05. Secondly, we have a CG that is smaller in number than the AFG. In this sense, we tried to make them as similar as possible in terms of age, sex, weight, and height. This may be a biasing factor that slightly modifies the characteristics observed between groups. On the other hand, whether the ankle fracture occurred on the dominant or non-dominant leg was not taken into account, which could be an effect-modifying factor. Finally, we only analysed dynamic plantar pressure at a global level. Future studies should include a regional analysis of the footprint during gait, at different stages of rehabilitation, in conjunction with other clinical-functional measures.

## 5. Conclusions

Patients with bimalleolar ankle fractures present clinical-functional deficits at 6 and 12 months after surgery that can be limiting in their daily life. During this time, atrophy of the calf musculature, increased ankle swelling, and a decrease in ADF ROM when comparing both limbs (OA/NOA) are observed. Likewise, a decrease in cadence and gait speed is evident, as well as moderate differences in dynamic plantar pressure when compared to healthy subjects. Despite the slight improvement between 6 and 12 months in ADF ROM and gait parameters, other measures such as bimalleolar/calf circumference and plantar pressures hardly changed. In addition, regression analysis revealed that the clinical-functional status of this population can condition between 30% and 67% of the result obtained in dynamic plantar pressure. Finally, despite the functional improvement one year after surgery, the AOFAS and OMAS scales did not show sufficiently high values to be considered a complete recovery.

## Figures and Tables

**Table 1 sensors-23-03975-t001:** Demographic, anthropometric, clinical, and functional characteristics of patients with bimalleolar ankle fractures 6 months after surgery and the control group.

Type	AFG (n = 22) Mean ± SD	95% CI	CG (n = 11)	95% CI
			Mean ± SD	
Age (years)	43.50 ± 10.20	39.0; 48.0	39.90 ± 8.60	34.10; 45.70
Sex Women (%); Men (%)	45% (W); 55% (M)		55% (W); 45% (M)	
Height (cm)	169.30 ± 9.50	164.80; 173.70	170.50 ± 7.90	165.20; 175.80
Weight (kg)	77.80 ± 10.60	73.10; 82.50	74.00 ± 9.10	67.90; 80.10
Limb Length (OA)/dominant limb CG * (cm)	85.60 ± 5.90	82.90; 88.20	86.20 ± 5.50 *	82.60; 89.90 *
Limb Length (NOA)(cm)	85.60 ± 5.90	82.90; 88.20		
Days from injury to surgery	4.80 ± 7.60	1.40; 8.10		
Immobilization (weeks)	3.40 ± 1.20	2.80; 3.90		
Rehabilitation time (weeks)	13.00 ± 2.40	11.90; 14.10		
AOFAS Ankle-Hindfoot score	73.60 ± 11.40	71.90; 75.30		
OMAS	57.30 ± 22.00	54.10; 60.60		
Corticosteroid use (% no)	95			
Arteriopathy (% no)	100			
Diabetes (% no)	100			
Complications (% no)	95			

AFG: Ankle Fracture Group; CG: Control Group; SD: Standard Deviation; CI: Confidence Interval; AOFAS: American Orthopaedic Foot and Ankle Society; OMAS: Olerud–Molander Ankle Score.

**Table 2 sensors-23-03975-t002:** Intra-group difference (AFG) between operated ankle/non-operated ankle and between groups (AFG-CG) of ankle perimeters (bimalleolar and calf) and ADF ROM at 6 months post-surgery.

	AFG (n = 22)		MD (OA-NOA)	CG (n = 11)	MD (AFG ^1^-CG)
	OA	NOA	(95% CI)	DL	(95% CI)
	Mean ± SD	Mean ± SD		Mean ± SD	
Calf perimeter (cm)	34.20 ± 4.00	35.50 ± 4.40	−1.30 (−2.00; −0.50) **	33.70 ± 2.50	0.50 (3.10; −2.30)
Bimalleolar perimeter (cm)	25.10 ± 2.10	24.10 ± 2.10	1.00 (0.80; 1.20) **	21.90 ± 1.60	3.20 (4.60; 1.70) *
ADF ROM (degrees)	22.80 ± 7.40	35.40 ± 5.30	−12.70 (−15.10; −10.30) **	41.90 ± 6.10	−19.10 (−13.80; −24.40) *

AFG: Ankle Fracture Group; CG: Control Group; OA: Operated Ankle; NOA: Non-operated Ankle; DL: Dominant Limb; SD: Standard deviation; CI: Confidence interval; ADF ROM: Ankle dorsiflexion range of motion; MD: Mean difference; ^1^ OA in AFG; * Significance level *p* < 0.05; ** Significance level *p* < 0.001.

**Table 3 sensors-23-03975-t003:** Intra-group difference (AFG) between operated ankle/non-operated ankle and between groups (AFG-CG) of gait parameters and dynamic plantar pressure at 6 months post-surgery.

	AFG (n = 22)		MD (OA-NOA)	CG (n = 11)	MD (AFG ^1^-CG)
	OA	NOA	(95% CI)	DL	(95% CI)
	Mean ± SD	Mean ± SD		Mean ± SD	
Cadence (steps/min)	99.90 ± 9.80			113.70 ± 5.20	−13.80 (−8.40; −19.10) **
Speed (m/s)	0.94 ± 0.10			1.18 ± 0.20	−0.24 (−0.12; −0.36) **
Peak plantar pressure (kPa/Kg)	1.28 ± 0.26	1.33 ± 0.25	−0.04 (−0.08; −0.01) *	1.60 ± 0.23	−0.32 (−0.50; −0.12) *
Average plantar pressure (kPa/Kg)	0.70 ± 0.14	0.72 ± 0.15	−0.02 (−0.03; −0.01) *	0.86 ± 0.11	−0.16 (−0.25; −0.06) *
Contact time (ms)	822.00 ± 136.00	846.80 ± 127.40	−24.80 (−39.90; −9.60) *	693.20 ± 127.40	128.80 (58.30; 199.10) **

AFG: Ankle Fracture Group; CG: Control Group; OA: Operated Ankle; NOA: Non-operated Ankle; DL: Dominant Limb; SD: Standard deviation; CI: Confidence interval; ADF ROM: Ankle dorsiflexion range of motion; MD: Mean difference; ^1^ OA in AFG; * Significance level *p* < 0.05; ** Significance level *p* < 0.001.

**Table 4 sensors-23-03975-t004:** Intra-group difference (AFG) between operated ankle/non-operated ankle and between groups (AFG-CG) of ankle circumferences (bimalleolar and calf) and ADF ROM at 12 months post-surgery.

	AFG (n = 21)		MD (OA-NOA)	CG (n = 11)	MD (AFG ^1^-CG)
	OA	NOA	(95% CI)	DL	(95% CI)
	Mean ± SD	Mean ± SD		Mean ± SD	
Calf circumference (cm)	34.80 ± 4.60	35.70 ± 4.30	−0.90 (−2.10; −0.60) **	33.70 ± 2.50	1.10 (−3.60; 1.40)
Bimalleolar circumference (cm)	24.90 ± 2.10	24.00 ± 2.10	0.90 (0.30; 1.40) *	21.90 ± 1.60	3.00 (1.40; 4.40) **
ADF ROM (degrees)	29.60 ± 9.10	37.10 ± 6.10	−7.40 (−4.30; −10.60) **	41.90 ± 6.10	−12.20 (−5.90; −18.50) **

AFG: Ankle Fracture Group; CG: Control Group; OA: Operated Ankle; NOA: Non-operated Ankle; DL: Dominant Limb; SD: Standard deviation; CI: Confidence interval; ADF ROM: Ankle dorsiflexion range of motion; MD: Mean difference; ^1^ OA in AFG; * Significance level *p* < 0.05; ** Significance level *p* < 0.001.

**Table 5 sensors-23-03975-t005:** Intra-group difference (AFG) between operated ankle/non-operated ankle and between groups (AFG-CG) of gait parameters and dynamic plantar pressure at 12 months post-surgery.

	AFG (n = 21)		MD (OA-NOA)	CG (n = 11)	MD (AFG ^1^-CG)
	OA	NOA	(95% CI)	DL	(95% CI)
	Mean ± SD	Mean ± SD		Mean ± SD	
Cadence (steps/min)	106.40 ± 7.60			113.70 ± 5.20	−7.30 (−1.90; −12.50) *
Speed (m/s)	1.01 ± 0.10			1.18 ± 0.20	−0.17 (−0.04; −0.28) *
Peak plantar pressure (kPa/Kg)	1.33 ± 0.26	1.35 ± 0.27	−0.02 (−0.04; 0.01)	1.60 ± 0.24	−0.27 (−0.45; −0.08) *
Average plantar pressure (kPa/Kg)	0.71 ± 0.15	0.72 ± 0.15	−0.01 (−0.02; 0.01)	0.86 ± 0.11	−0.15 (−0.25; −0.04) *
Contact time (ms)	765.80 ± 82.30	768.00 ± 89.70	−2.60 (−9.70; 0.40)	693.20 ± 127.40	72.60 (14.60; 130.50) *

AFG: Ankle Fracture Group; CG: Control Group; OA: Operated Ankle; NOA: Non-operated Ankle; DL: Dominant Limb; SD: Standard deviation; CI: Confidence interval; ADF ROM: Ankle dorsiflexion range of motion; MD: Mean difference; ^1^ OA in AFG; * Significance level *p* < 0.05.

**Table 6 sensors-23-03975-t006:** Intra-group difference (AFG) at 6 and 12 months of clinical measures, gait parameters, dynamic plantar pressure, and functional scales.

		AFG 6 m (n = 21)	AFG 12 m (n = 21)	MD
		Mean ± SD	Mean ± SD	(95% CI)
Clinical measures ^1^	Calf circumference (cm)	34.20 ± 4.00	34.80 ± 4.60	−0.6 (−1.40; 0.20)
	Bimalleolar circumference (cm)	25.10 ± 2.10	24.90 ± 2.10	0.2 (−0.40; 0.70)
	ADF ROM (degrees)	22.80 ± 7.40	29.60 ± 9.10	−6.80 (−10.90; −2.50) *
Gait parameters	Cadence (steps/min)	99.90 ± 9.80	106.40 ± 7.60	−6.50 (−10.20; −2.30) *
	Speed (m/s)	0.94 ± 0.10	1.01 ± 0.10	−0.07 (−0.11; −0.040) **
Dynamic plantar pressure ^1^	Peak plantar pressure (kPa/Kg)	1.29 ± 0.27	1.34 ± 0.26	−0.05 (−0.08; −0.01) *
	Average plantar pressure (kPa/Kg)	0.70 ± 0.14	0.71 ± 0.15	−0.01 (−0.02; 0.01)
	Contact time (ms)	822.00 ± 136.00	765.80 ± 82.30	56.20 (9.90; 99.50) *
Functional scales	AOFAS Ankle-Hindfoot score	73.60 ± 11.50	84.40 ± 12.40	10.80 (7.10; 14.50) **
	OMAS	57.30 ± 22.00	80.00 ± 25.00	22.70 (15.30; 29.90) **

SD: Standard deviation; CI: Confidence interval; ADF ROM: Ankle dorsiflexion range of motion; MD: Mean difference; ^1^ OA in AFG; * Significance level *p* < 0.05; ** Significance level *p* < 0.001.

**Table 7 sensors-23-03975-t007:** Correlation between clinical measurements, gait parameters, and functional scales with dynamic plantar pressure in operated ankle at 6 and 12 months after surgery.

			Clinical Measurements, Gait Parameters, and Functional Scales						
	Dynamic Plantar Pressure	Age	ADF ROM	Bimalleolar Circumference	Calf Circumference	Cadence	Speed	AOFAS	OMAS
6 months after surgery	Peak plantar pressure (kPa/Kg)	0.64	−0.28	−0.58 *	−0.45 *	0.21	0.14	−0.13	−0.11
	Average plantar pressure (kPa/Kg)	0.18	0.23	−0.52 *	−0.45 *	0.23	0.18	−0.23	−0.27
	Contact time (ms)	0.18	−0.55 **	−0.32	−0.01	−0.81 **	−0.81 **	−0.45 *	−0.4
12 months after surgery	Peak plantar pressure (kPa/Kg)	−0.05	0.29	−0.67 **	−0.61 **	0.17	0.15	−0.23	−0.16
	Average plantar pressure (kPa/Kg)	0.06	0.19	−0.59 **	−0.52 *	0.27	0.14	−0.08	−0.19
	Contact time (ms)	0.21	−0.26	−0.37	0.15	−0.68 **	−0.67 **	−0.32	−0.18

Pearson’s correlations (*r*); * *p* < 0.05; ** *p* < 0.001.

## Data Availability

Not applicable.

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
