# Peer review of "Analysis of Dynamic Plantar Pressure and Influence of Clinical-Functional Measures on Their Performance in Subjects with Bimalleolar Ankle Fracture at 6 and 12 Months Post-Surgery"

_sensors, 2023, doi:10.3390/s23083975_

Round 1

Reviewer 1 Report

The study by Fernandez-Gorgojo et al is really interesting and analyzes in depth the biomechanical and functional alterations of patients after ankle fracture. Although it shows a small sample size, the differences found within and between groups are relevant and provide information of interest for the management and follow-up of patients at 6 and 12 months after surgery. I agree with the authors that a more detailed analysis of plantar pressures (e. g.,  dividing them into the three rockers of gait) would be of great interest to identify the phase of stance most affected by the alterations described in the current work.

As far as I can recall, the topic is relevant and provides valuable information for the assessment and management of patients with ankle fractures. Its strength is precisely the differences found between subjects with and without pathology for different biomechanical variables that are relatively easy to measure. In addition, it was found that the alterations persist beyond twelve months after the intervention, which has direct practical application for clinicians and rehabilitators.

On the other hand, there was quite a lot of information on kinematic or other variables such as ankle function or dorsiflexion ranges of motion, and perhaps some additional assessment of muscle strength or balance was missing.

Given that it is an extensive work that includes a fairly comprehensive assessment of patients after ankle fracture, with important clinical implications and an appropriate use of biomechanical assessment sensors in the healthcare setting.

Minor comments:

Line 131: please check if there is a missing citation or delete a period

Line 380: the sentence should be rewritten so as not to repeat "in contrast" twice

Reviewer 2 Report

I received papers for review entitled: Analysis of dynamic plantar pressure and influence of clinical functional measures on their performance in subjects with bimalleolar ankle fracture at 6 and 12 months post surgery This is an interesting study, but I have a few comments.

The group of 22 subjects is a small group. Especially considering both genders and the large age range of the respondents. Has age been taken into account in the analysis? Different Regeneration Abilities are in 18 and 50 year old person   Did the researchers take into account co-morbidities and other health and/or musculoskeletal collisions? One year is quite a long time.   Table 2 is duplicated and incomplete

Reviewer 3 Report

1- abstract  can be improved by answer the three classic question ,why do this work, how doing it, what the important results 

2- The authors must compared his results with others works

3- conclusion needs to rewritten in order to showing the basic conclusion points

Reviewer 4 Report

Well written manuscript with clear and detailed description of the work carried being out. Inclusion and exclusion criteria were clearly defined.

I notice several minor mistakes:
1. Line 377- spelling mistake
2. Line 131- typing mistake
3. Line 177- subjects should be named he/ she because there were female subjects as well
4. Table 2 and Table 5 were repeated twice, with the repeated ones have no data.
5. Caption for table 4 and table 5- add word ‘at” between ADF ROM and 12 months

I would like to know more on these following:
1. Line 192- which author controlled/ decided to eliminate incorrect/ incomplete steps? You may indicate the author (initials) in the text.
2. Line 238- no data from 1 subject at 12 months- didn’t it affect power analysis for data at 12 months? why didn’t authors exclude out data from this 1 subject at 6 months?
3. The authors suggested the plantar pressure should be analysed, however this is only possible to be done with pressure platform available, right? It should be read by 1 assessor only for consistency, right? Seems like not so practical. In addition, Table 6 showed not much difference in plantar pressure between 6 months and 12 months compared to other parameters, therefore it implies plantar pressure is not sensitive parameter to indicate recovery from ankle fracture.
4. Table 1- what is the reason for presenting 95%CI
5. I am still puzzled with the need for including the CG subjects in this study. Could the authors explain more?
6. Measured DL in CG, but in AFG, there was no record of OA occurred on DL or not. DL can be confounding factor on some parameters eg circumference. Especially when considering the finding indicating dynamic plantar pressure significantly different between OA and CG, it is possible that whether ankle fracture happened to to dominant leg or not could be a confounding factor.
7. Table 2 and Table 5- why data for CG is Media +/- DE and not Mean +/- SD?
8. I would like to suggest the authors to improve the conclusion; whether the data answered the research question or not.
